# Predictive Modeling for Spinal Metastatic Disease

**DOI:** 10.3390/diagnostics14090962

**Published:** 2024-05-05

**Authors:** Akash A. Shah, Joseph H. Schwab

**Affiliations:** 1Department of Orthopaedic Surgery, David Geffen School of Medicine at UCLA, Los Angeles, CA 90095, USA; 2Department of Orthopaedic Surgery, Cedars-Sinai Medical Center, Los Angeles, CA 90048, USA; joseph.schwab@cshs.org

**Keywords:** machine learning, spinal metastasis, orthopedic oncology, oncology

## Abstract

Spinal metastasis is exceedingly common in patients with cancer and its prevalence is expected to increase. Surgical management of symptomatic spinal metastasis is indicated for pain relief, preservation or restoration of neurologic function, and mechanical stability. The overall prognosis is a major driver of treatment decisions; however, clinicians’ ability to accurately predict survival is limited. In this narrative review, we first discuss the NOMS decision framework used to guide decision making in the treatment of patients with spinal metastasis. Given that decision making hinges on prognosis, multiple scoring systems have been developed over the last three decades to predict survival in patients with spinal metastasis; these systems have largely been developed using expert opinions or regression modeling. Although these tools have provided significant advances in our ability to predict prognosis, their utility is limited by the relative lack of patient-specific survival probability. Machine learning models have been developed in recent years to close this gap. Employing a greater number of features compared to models developed with conventional statistics, machine learning algorithms have been reported to predict 30-day, 6-week, 90-day, and 1-year mortality in spinal metastatic disease with excellent discrimination. These models are well calibrated and have been externally validated with domestic and international independent cohorts. Despite hypothesized and realized limitations, the role of machine learning methodology in predicting outcomes in spinal metastatic disease is likely to grow.

## 1. Introduction

The spinal column is the most common site of bony metastatic disease [1]. An estimated 40–70% of all patients with cancer develop spinal metastasis, with 5–10% developing metastatic epidural compression [1,2]. As the survival rates of most primary malignancies increase, the prevalence of symptomatic spinal metastatic disease is expected to increase. Patients with symptomatic spinal metastases may present with pain, radicular symptoms, mechanical instability, or neurologic decline.

The treatment of spinal metastases is largely palliative, with a focus on pain relief, preserving or restoring neurologic function, local oncologic control, and maintaining mechanical stability of the spinal column. Spinal metastases were historically treated with conventional external beam radiation therapy with or without corticosteroids. Patients with radiosensitive malignancies (e.g., multiple myeloma and prostate cancer) responded well, while relatively radioresistant malignancies did not (e.g., sarcoma and colorectal cancer) [3,4,5]. In 2005, Patchell and colleagues reported a randomized controlled trial comparing surgical decompression and radiation therapy in patients with metastatic epidural compression. They found that surgical decompression combined with radiation yielded improved survival, maintenance and recovery of ambulation, decreased opioid requirements, and improved neurologic outcomes compared to radiation therapy alone [6]. The rate of surgical management of spinal metastatic disease has increased, with evidence that palliative surgery is valued by patients and their families [7,8].

Surgical treatment involves decompression and/or instrumented fusion. There is a wide spectrum of invasiveness that surgical decompression may entail, ranging from minimally invasive laminotomies to open laminectomy and even corpectomy. Similarly, instrumented fusion may comprise open posterolateral fusion or percutaneous pedicle screw fixation. Surgical management carries with it considerable perioperative morbidity including incidental durotomy, wound infection, and venous thromboembolism [9]. It is thus of great utility to determine a patient’s overall prognosis before deciding on a specific treatment strategy.

In this narrative review, we aim to describe the tools that have been developed to predict mortality in patients with extramedullary spinal metastasis. While there have been numerous studies reviewing spinal oncology and artificial intelligence in spinal surgery, we provide a focus on conventional and machine learning-driven methods for modeling mortality in spinal metastasis specifically. We first explain the NOMS decision framework that organizes the assessment of a patient with spinal metastatic disease into four categories. This framework hinges on the patient’s prognosis. We detail the multiple prognostic systems proposed for predicting mortality in spinal metastatic disease; these have been largely developed with features derived from expert opinion or logistic regression. As machine learning methods have been utilized in spinal surgery with great success, machine learning-driven models have been built to predict mortality in spinal metastatic disease. We describe the development and validation of these models. While these models represent an important advance in our ability to predict valuable prognostic information for patients with spinal metastasis, they carry specific risks and limitations. Finally, we discuss future directions for how machine learning and artificial intelligence tools can continue to improve our ability to predict mortality in patients with spinal metastasis.

## 2. NOMS Framework

The NOMS decision framework was developed to guide decision making for the treatment of symptomatic spinal metastatic disease. It distills the assessment of spinal metastases into four considerations: neurologic function, oncologic control, mechanical stability, and systemic disease burden [10].

The neurologic considerations assess both clinical and imaging findings. Clinical findings include myelopathy and functional radiculopathy. Bilsky and colleagues report a validated magnetic resonance-based scoring system to evaluate the degree of epidural spinal cord compression. Utilizing axial T2-weighted images, the extent of epidural compression is graded into low-grade lesions without cord compression (grades 0 and 1a–c) or higher grade lesions with an increasing degree of cord compression (grades 2 and 3) [11]. The oncologic consideration is based on the expected clinical response to radiation and/or systemic therapy. Primary tumor histology is the most important predictor of response to conventional external beam radiation therapy. Patients with radiosensitive malignancies may be treated with conventional radiation therapy regardless of the degree of epidural compression, whereas those with less favorable primary tumor histology may require stereotactic radiosurgery with or without surgical decompression [2,10,11].

Mechanical instability of the spine is an indication for surgery regardless of the neurologic or oncologic assessments. Incorporating both clinical and radiologic features, the Spinal Instability Neoplastic Score (SINS) is a validated scoring system that was devised to identify patients with instability that may require surgical stabilization (e.g., posterolateral fusion and percutaneous pedicle screw fixation). The SINS comprises six parameters: location, mechanical pain, lytic versus blastic lesion, spinal alignment, vertebral body collapse, and involvement of the posterior elements. Patients with a low SINS (0–6) may not require surgical stabilization, whereas patients with a high SINS (13–18) would benefit from stabilization [12,13].

Perhaps the most important consideration for treatment is whether the patient can tolerate a proposed intervention. This is based on the patient’s systemic medical comorbidities and overall tumor burden. The neurologic, oncologic, and mechanical considerations are of significance; however, the patient should survive long enough to make perioperative morbidity and risks worthwhile. Similarly, we do not want to withhold treatment from a patient who would benefit from it.

There is significant variability between the complication profiles of radiation therapy, minimally invasive surgery, and open decompression and fusion. Surgery for spinal metastasis carries with it considerable perioperative morbidity; complications include wound infection, pain, neurologic deterioration, and venous thromboembolism [14,15,16]. Accurate prognosis is thus crucial when determining which treatment modalities carry an acceptable risk–benefit ratio and are in line with the patient’s goals of care. Providing accurate estimates of mortality would be of great utility to patients and families, improving informed decision making.

## 3. Prognostic Scoring Systems

### 3.1. Modeling Approaches

The goal of a predictive model is to determine the relationship between existing data and a future outcome. Most prognostic models predicting survival for spinal metastatic disease have done so with regression techniques (e.g., logistic regression). These models employ methods that are grounded in statistical inference. As an example, statistical inference may be employed to identify risk factors independently associated with one-year mortality; the odds ratios of these factors would then be weighted and used to build an equation that determines the risk of mortality. Such models are optimized for interpretability—that is, identifying which features drive mortality. In contrast, machine learning models are optimized for prediction. Spanning a vast array of learning algorithms (e.g., gradient boosting and Bayes Point Machine), machine learning models interrogate relationships between multiple variables in large datasets with the purpose of maximizing predictive performance. They often utilize complex equations to achieve this end, limiting their interpretability [17].

### 3.2. Classical or Regression-Driven Scores

Despite the recognition that patient survival is a key consideration in the planning of treatment for spinal metastatic disease, physicians have historically relied on clinical judgment to predict prognosis. Survival predictions by oncologists, radiation oncologists, and surgeons have been consistently determined to be inaccurate and overly optimistic [18,19,20,21,22]. Spine surgeons tend to overestimate life expectancy, leading to invasive procedures with greater associated perioperative morbidity [22]. There has thus been a significant effort to predict prognosis in patients with spinal metastatic disease in a data-driven manner with greater accuracy. 

Tokuhashi and colleagues published a series of 64 patients with operatively treated spinal metastatic disease, utilizing 6 preoperative features to develop a scoring system to determine survival: performance status, the number of extraspinal bony metastases, the number of spinal metastases, the presence of visceral metastasis, primary tumor histology, and neurologic deficit. Out of a total score of 12, patients with a score ≤ 5 survived 3 months or less on average, whereas those with a score ≥ 9 survived 12 months or more [23]. They recommend that patients with a score ≤ 5 undergo palliative treatment, while those scoring ≥ 9 undergo surgical excision. In 2005, the authors reported an updated scoring system where they utilized the same 6 features in 246 patients composed of those treated conservatively and operatively; however, they expanded the primary tumor histology score to 0–5, hoping to provide more prognostic value to lesion biology. Out of a total score of 15, a score ≤ 8 is associated with survival < 6 months, with the authors recommending conservative treatment. A score between 9 and 11 is associated with survival > 6 months, with recommended palliative surgery. A score ≥ 12 is associated with survival > 1 year and excisional surgery is recommended by the authors [24].

In 1995, Bauer and Wedin assessed survival after surgery in 153 patients with extremity metastases and 88 with spinal metastases. They found five positive prognostic factors for 1-year survival: the absence of visceral metastasis, the absence of pathologic fracture, solitary spinal metastasis, no lung cancer, and primary tumor histology of breast, kidney, lymphoma, or multiple myeloma. The greater the number of positive prognostic factors, the higher the probability of 1-year survival [25]. Leithner and colleagues proposed a modified Bauer score in which they removed pathologic fracture from the list of prognostic features [26]. For the modified Bauer score, the authors recommend non-operative management, dorsal palliative surgery, or ventral–dorsal excisional surgery depending on the number of positive prognostic factors. 

Spurred by the growing popularity of wide excisional resection for spinal metastases, Tomita and colleagues developed a prognostic scoring system for patients to guide surgical strategy. They included primary tumor histology, the presence of visceral metastasis, and the number of spinal/extraspinal bony metastases to generate a prognostic score ranging from 2 to 10. The lower the score, the more aggressive the recommended surgical strategy. A score of 2 or 3 was associated with a recommendation of wide or marginal excision, with the treatment goal of long-term local control, while a score of 4 or 5 yielded a recommendation of marginal or intralesional excision, with the goal of middle-term local control. A score of 6 or 7 led to a recommendation of palliative surgery, with the treatment goal of short-term palliation. Patients with scores ≥ 8 were recommended to undergo supportive care [27].

In 2005, van der Linden and colleagues published a prospectively randomized radiation therapy trial including 342 patients with symptomatic spinal metastases without neurologic deficit. They proposed a scoring system based on three prognostic factors: performance status, primary tumor histology, and the presence of visceral metastasis. Patients with ≤3 out of 6 points had a median survival of 3 months, 4 or 5 points was associated with a median survival of 9 months, and 6 points was associated with a median survival of 18.7 months [28].

Katagiri and colleagues reported a prognostic scoring system in 2005 for patients with skeletal metastasis who underwent non-operative or operative management. Similar to Bauer and Wedin, their cohort of 350 patients included both spinal and extremity lesions. They identify performance status, multiple skeletal metastases, the presence of visceral metastasis, and prior systemic therapy as important prognostic factors in their scoring system out of 8 points [29]. In 2014, they published a revised score with three major changes. They modified the primary histology breakdown to slow-growth, moderate-growth, and rapid-growth lesions based on the molecular characterization of the primary lesion (e.g., hormone-dependent versus hormone-independent lung cancer with or without molecularly targeted drugs) and refined the visceral metastasis factor as nodular/cerebral metastasis or disseminated metastasis. Additionally, they included laboratory data (e.g., C-reactive protein and serum albumin). The higher the number of points out of 10, the lower the survival [30].

With 307 patients who underwent surgery for spinal metastasis across 4 centers, Ghori and colleagues developed the New England Spinal Metastasis Score (NESMS). Feeling that the modified Bauer score inadequately assessed the general health status of patients, the authors included ambulatory status and serum albumin levels in addition to the modified Bauer score in their scoring system. The NESMS was externally validated on a cohort of operatively managed patients as well as separately validated on a cohort of non-operatively managed patients [31,32]. Additionally, a prospective analysis found that the NESMS shows superior predictive capability for 1-year mortality compared to the Tokuhashi, Tomita, and SINS systems [33].

In 2016, the SORG Orthopaedic Research Group published a survival algorithm for patients with operatively managed spinal metastatic disease. With 649 patients, they identified the following prognostic factors: age ≥65 years, performance status, primary tumor histology, multiple spinal metastases, visceral metastasis, prior systemic therapy, white blood cell count ≥ 11,000/μL, and hemoglobin ≤ 10 g/dL. Based on the number of points out of 12, the authors divided patients into good, intermediate, and poor prognosis groups, with associated survival probabilities of 30, 90, and 365 days. To provide a more user-friendly tool for clinicians, they additionally built a nomogram that converts each of the aforementioned factors into a score. The points are summed and translated into a patient-specific risk of survival at 30, 90, and 365 days [34]. The SORG nomogram was externally validated for 90-day and 1-year mortality on an independent cohort of 100 patients, outperforming the Tokuhashi score, Tomita score, modified Bauer score, and NESMS [35]. The nomogram was additionally externally validated for 30-day and 90-day survival after surgery, outperforming eight other scoring systems [36]. A recent systematic review similarly found that the SORG nomogram performs the best out of 17 scoring systems at predicting 90-day and 1-year mortality [37]. A summary of the discussed prognostic scoring systems is provided in Table 1.

Primary tumor histology and the presence of visceral lesions were most frequently identified as being important for prognosis; they were included in 8 of the 10 aforementioned studies. Pretreatment performance status was included in six studies. Table 2 details the most commonly included features in conventional prognostic models.

### 3.3. Machine Learning-Driven Scoring Systems

The original Tokuhashi, revised Tokuhashi, and Tomita studies do not report utilizing statistical modeling to derive the prognostic factors that they included in their scoring systems [23,24,27]. The remaining prognostic scoring systems between 1990 and 2016 for predicting survival in patients with spinal metastatic disease were largely developed with Cox proportional hazards or multivariable logistic regression to identify prognostic factors and predict short- or long-term mortality [25,26,28,29,30,34,38]. Although regression techniques are often successful in assessing the association between explanatory features and outcomes, they are less suited for prediction. Machine learning methods have immensely grown in popularity in the past decade due to their ability to better capture complex non-linear relationships and factor–factor interactions compared to traditional regression techniques [39,40]. Machine learning algorithms have been shown to successfully predict clinical outcomes across diverse clinical disciplines, often outperforming logistic regression [41,42,43]. Additionally, machine learning modeling allows for the assessment of model discrimination, model calibration, and decision curve analysis—important metrics for determining the utility of a reported model [44].

The SORG Orthopaedic Research Group performed the first published machine learning-driven analysis for the prediction of survival after operatively managed spinal metastasis in 2016. They developed a boosting algorithm that performed better at all time points for the prediction of 30-day, 90-day, and 1-year survival in the training dataset but performed slightly worse than the nomogram in the testing dataset [34]. They constructed the model with factors shown to be independently associated with survival: age ≥65 years, performance status, primary tumor histology, multiple spinal metastases, visceral metastasis, prior systemic therapy, white blood cell count ≥ 11,000/μL, and hemoglobin ≤ 10 g/dL.

With 1790 patients from the American College of Surgeons National Surgical Quality Improvement Program, the SORG group built a Bayes Point Machine algorithm for the prediction of 30-day mortality after surgery for spinal metastasis. This machine learning algorithm performed well, with an area under the receiver-operating characteristic curve (AUROC) of 0.782, and was well calibrated. Preoperative patient features used for algorithm development included serum albumin level, performance status, white blood cell count, hematocrit, alkaline phosphatase level, American Society of Anesthesiologists class, and location within the spine. The Bayes Point Machine was incorporated into a web application and deployed as an open-access tool for clinicians, the first such web-based application for the prediction of survival after spinal metastasis [45].

The SORG group then transitioned to building machine learning-driven models with high-quality institutional cohorts. With 732 patients from 2 large academic medical centers in Massachusetts, Karhade and colleagues developed stochastic gradient boosting algorithms that accurately predicted 90-day and 1-year mortality for operatively managed spinal metastatic disease. Features important for both models included serum albumin level, primary tumor histology, functional status, absolute lymphocyte count, alkaline phosphatase level, and creatinine level. Both models were well calibrated with excellent discrimination, with AUROC values of 0.83 and 0.89 for 90-day mortality and 1-year mortality, respectively. The discriminations of the stochastic gradient boosting models for 90-day and 1-year mortality were greater than those of the following scores: original Tokuhashi, revised Tokuhashi, Tomita, modified Bauer, van der Linden, Katagiri, NESMS, and SORG nomogram. The authors also reported a web-based calculator where users input patient feature values and receive a probability of 90-day or 1-year mortality [46]. These models were externally validated multiple times, first with a single independent cohort spanning patients from 2003 and 2016 in a tertiary-care medical center in Maryland [47]. They were externally validated again by Bongers and colleagues with a contemporary cohort composed of 200 patients from a tertiary-care center in New York between 2014 and 2016, performing excellently and with good calibration [48]. Despite taking important steps toward the generalizability of these models, both of these external validation studies were performed in the same geographic region of the United States that the SORG algorithms were developed in. Shah and colleagues successfully externally validated these algorithms with a geographically distinct cohort from a tertiary-care center in California. Furthermore, they showed that the algorithms perform accurately in a contemporary cohort subgroup treated in 2015 or later [49]. Additionally, Yang and colleagues reported successful international external validation of the SORG algorithms with a Taiwanese cohort of 427 patients [50]. Zhong and colleagues also reported successful external validation of the SORG algorithms in a cohort of 150 patients with lung-derived spinal metastasis [51].

It has been postulated that a patient is likely to benefit from surgical intervention for spinal metastatic disease if he/she is likely to live longer than 3 months [6,52]. Three months has thus been a survival time point that most of the classical and machine learning models have aimed to predict. With the advent of minimally invasive techniques that may be associated with less perioperative morbidity, the survival required to benefit from surgery may be less than this original cut-off. The SORG group thus sought to predict survival at an earlier time point. With an institutional cohort of 3001 patients with spinal metastasis treated non-operatively or operatively, they developed a machine learning model predicting 6-week mortality. The elastic net penalized logistic regression model was well calibrated, with excellent discrimination (AUROC: 0.85). They found that the most important features for prediction were serum albumin level, primary tumor histology, absolute lymphocyte count, the number of spinal metastases, and functional status. This algorithm was externally validated with 1303 patients from 4 independent external cohorts from the United States and Taiwan [53]. A summary of the machine learning studies for spinal metastatic disease is provided in Table 3.

Performance status was the most commonly employed feature in the SORG algorithms, included in all four models. The frequency of features viewed as important for the machine learning-driven models are detailed in Table 4.

Many of the same features are included in the prognostic models built with conventional statistics and those built with machine learning algorithms: primary tumor histology, the presence of visceral metastases, the number of spinal metastases, serum albumin level, white blood cell count, hemoglobin level, and prior systemic therapy. The SORG machine learning models did not include the number of bony metastases, neurologic symptoms, or the presence of pathologic fracture. Additionally, both modeling approaches employed discrete features; there were no included continuous variables or imaging variables. The machine learning models included a greater number of features on average per model compared to the conventional models (8.8 features versus 4.7 features), primarily due to an increased number of laboratory values included. Earlier models built with traditional statistics attempted to distil the complex interplay of features that influence prognosis into a small number of features, with the aim of simplifying the features that physicians had to consult to receive accurate and interpretable information about prognosis. In contrast, a greater number of features tend to optimize the performance of machine learning models. The higher dimensionality of inputs maximizes predictive performance, often at the expense of interpretability.

Machine learning-driven analyses to predict survival for spinal metastatic disease represent a major advance in the ability of clinicians to accurately assess prognosis. These algorithms perform very well, with excellent discrimination, and are well calibrated across multiple time points—successfully predicting survival at 30 days, 6 weeks, 90 days, and 1 year. By quantifying risk with more granularity than simply “low-risk” or “high-risk”, these tools can facilitate a data-driven discussion of risks and benefits, with subsequently improved preoperative patient counseling. Additionally, the machine learning analyses were shown to be internally valid, with the reporting of key metrics for the validation of clinical prediction models: model discrimination, model calibration, and decision curve analysis. Reporting these metrics is crucial to adequately assess the clinical utility of any predictive algorithm. Finally, the lack of external validation is a pervasive problem in the machine learning literature—significantly limiting the generalizability of these models [54]. The SORG machine learning models have been externally validated on multiple occasions with contemporary, geographically distinct domestic and international cohorts. The classical regression-based studies generate a total score based on the presence or absence of prognostic factors and recommend treatment based on which prognostic group the patient falls into. The SORG machine learning algorithms from 2019 onward provide web-based risk calculators that generate the patient-specific probability of mortality. These risk calculators allow clinicians to input features and receive the patient-specific probability of mortality as an output in a user-friendly digital interface.

Clinical data that are readily available in patient charts comprise the majority of input features used to develop these models. These include features such as primary tumor histology, the number of spinal metastases, the presence of visceral metastasis, and laboratory values. While primary tumor histology reflects the molecular characteristics of specific malignancies, molecular markers or gene expression data are yet to be incorporated into machine learning models for spinal metastasis mortality. Similarly, although deep learning methods have been applied to imaging for spinal metastasis, imaging data that are more detailed than the number of spinal metastases have not yet been included in prognostic models for spinal metastasis [55,56]. Additionally, patient-reported outcomes have not been utilized in reported models. Expanding the sources of input data to include molecular markers, imaging characteristics, and patient-reported outcomes is likely to improve predictive performance.

## 4. Assessing Machine Learning Algorithms

It is critical to determine the quality and validity of the proposed clinical prediction models. Let us consider internal validity and external validity separately. Internal validity refers to the validity of a model’s performance on the cohort for which it was developed (i.e., the derivation cohort). An important characteristic of a clinical prediction tool is not only how reproducible it is but also how applicable it is to an independent population. While internal validity is a measure of the reproducibility of a model, external validity is a measure of generalizability [57]. Despite being critical to establish the utility of prediction tools, external validation studies are rare in the machine learning literature [54].

Aiming to provide rigorous assessment of the validity of a predictive model, Steyerberg and Vergouwe proposed reporting and evaluating four metrics: discrimination, the calibration slope, the calibration intercept, and decision curve analysis [57]. An accurate model should reliably discriminate between those who develop an outcome and those who do not. Discrimination is typically measured with the AUROC, also known as the concordance statistic or c-statistic. The AUROC represents the probability that the model can distinguish between a patient who develops an outcome and one who does not. An AUROC of 1 represents perfect discrimination and an AUROC of 0.5 indicates random prediction by the model [57].

Good discriminatory capability is necessary but not sufficient for overall model performance. In addition to discriminating an event from a non-event, a model’s predictions should align with the observed outcomes within the study population. For every 100 patients for which a model predicts a 90% probability of 1-year survival, 90 patients should be alive in one year. Calibration captures this characteristic of a model; it is a measure of the agreement between the model predictions and the observed outcomes. Calibration is assessed with the calibration intercept and the calibration slope. The calibration intercept is a measure of the degree to which a model overestimates or underestimates the outcome of interest. A perfect model has a calibration intercept of 0. If a model predicting survival for spinal metastatic disease has a positive calibration intercept, it overestimates survival; an intercept value less than 0 means that the model underestimates survival. The calibration slope is a measure of how extreme a model’s predictions are [57,58]. A perfect model has a calibration slope of 1.

While discrimination and calibration assess the accuracy of a model’s predictions, they do not determine the clinical usefulness of a model. First introduced by Vickers and colleagues, decision curve analysis assesses clinical strategies by evaluating their net benefit across different probability thresholds [59]. The net benefit accounts for the potential benefit or harm from predictions issued by a model. It is a measure of the net true positives for a given threshold probability. A net benefit of 0.05 for a given threshold probability means that applying the model to 100 patients allows for the identification of 5 extra true positives without increasing the false positive rate [57,60]. The decision curve is drawn with the threshold probability on the x-axis and the net benefit on the y-axis. A model’s performance is compared to the default strategies of changing management for all or no patients [44]. The decision curve is thus a graphical depiction of the trade-off between true positive and false positive predictions for a model [60]. Decision curves for different models can be compared to determine which models maximize the net benefit. The decision curves for the SORG machine learning models predicting 90-day and 1-year mortality in an independent cohort of surgically treated patients with spinal metastasis are shown in Figure 1.

## 5. Future Directions and Limitations of Artificial Intelligence

The scope of artificial intelligence in orthopedic surgery has expanded considerably in recent years, growing to include both unsupervised learning and natural language processing algorithms. As clinical datasets grow in both number and dimensionality, machine learning-driven algorithms will likely be utilized even more rapidly in orthopedic surgery and spinal surgery [61,62,63].

Advances in the molecular and immunohistochemical characterization of malignancies have led to an increase in biological data that should be incorporated into machine learning algorithms. Additionally, imaging data from computed tomography and magnetic resonance imaging studies should be incorporated into prognostic models. Other disciplines of machine learning such as natural language processing and unsupervised learning models may improve the prediction of outcomes; these remain underutilized for the prediction of outcomes in spinal metastatic disease. The clinical utility of machine learning algorithms would be enhanced by incorporating them into the electronic health record (EHR). Although care must be taken to protect patient confidentiality, embedding machine learning-driven risk calculators into the EHR would allow for the automated collection of relevant data, with the ability to continually self-update with prospectively collected data.

Despite the successful published applications of machine learning methodology for spinal metastatic disease, there are multiple potential limitations that must be addressed as the uptake of this technology increases. Machine learning methods are optimized for prediction but not necessarily for explanation. The individual effects of factors employed by advanced machine learning methods on the outcome of interest are not straightforward to interpret in the way that regression coefficients for traditional statistical modeling are—leading to a “black box” phenomenon.

Additionally, machine learning methods are prone to overfitting the data on which they are developed. It is thus crucial that robust internal and external validation strategies be employed for all predictive algorithms, with standardized reporting of key validation metrics (i.e., discrimination, calibration, and decision curve analysis). Prospective testing of these models is key. Finally, machine learning models perform well due to their ability to detect patterns within a dataset. As a result, the quality of the data source determines the quality of the models built with those data. Missing or inaccurate data can severely impact the performance of machine learning models. Furthermore, machine learning models may have the unintended consequence of propagating biases. Existing disparities in cohort creation and treatment strategies may be exacerbated, potentially causing harm to underrepresented groups such as ethnic minorities or those of lower socioeconomic status [64].

The most significant barrier to the clinical implementation of prognostic models in spinal metastatic disease is the lack of prospective randomized trials that interrogate the effectiveness of machine learning-driven predictions. While successful external validation studies improve the argument for generalizability, prospective randomized trials remain the gold standard for proving that prognostic tools yield improved outcomes for patients. Prospective trials must be designed to test the reported models.

## 6. Conclusions

The considerable improvements in data science must continue to be applied to the management of spinal metastatic disease. Symptomatic spinal metastatic disease is an important and growing clinical entity that requires a data-driven approach. Prognosis is a fundamental data point that patients and their families rely on to determine the next step in management. Patients with spinal metastatic disease are typically older patients with multiple medical comorbidities; even with improvements in perioperative care, surgical intervention in these patients has a considerable complication profile. Setting realistic expectations is a crucial component appropriately informing the risk–benefit calculation that patients and their families must make when considering treatment options. Machine learning methods have built on the important work of classical prognostic algorithms, allowing for the accurate prediction of mortality in spinal metastatic disease. While these validated tools provide patient-specific risk of mortality, they do not and cannot tell clinicians how to treat a patient. It is not within the purview of models to determine treatment for a patient. How to treat symptomatic spinal metastasis is challenging and requires consideration of both clinical and psychosocial components. Shared decision making between patients, families, and their physicians is required to determine a treatment plan. Machine learning-driven algorithms can facilitate improved shared decision making by providing an accurate assessment of prognosis.

## Figures and Tables

**Figure 1 diagnostics-14-00962-f001:**
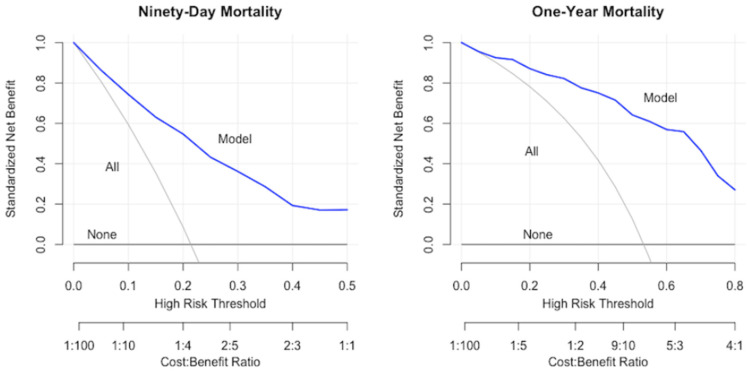
Decision curves for SORG models predicting 90-day and 1-year mortality after surgery for spinal metastasis. Figure obtained with permission from authors of Shah et al., 2021 [49].

**Table 1 diagnostics-14-00962-t001:** Conventional prognostic scoring systems.

Study	Number	Treatment	Location	Features	Points	Model
Tokuhashi et al., 1990 [23]	64	Operative	Spine	Performance status, the number of bony lesions, the number of spinal lesions, visceral lesions, primary histology, and neurologic deficit	12	None
Tokuhashi et al., 2005 [24]	264	Operative and non-operative	Spine	Performance status, the number of bony lesions, the number of spinal lesions, visceral lesions, primary histology, and neurologic deficit	15	None
Bauer and Wedin, 1995 [25]	241	Operative	Spine and extraspinal region	The number of spinal lesions, visceral lesions, primary histology, and pathologic fracture	5	Multivariate logistic regression
Leithner et al., 2008 [26]	69	Operative	Spine	The number of spinal lesions, visceral lesions, and primary histology	4	Multivariate logistic regression
Tomita et al., 2001 [27]	67	Operative and non-operative	Spine	The number of bony lesions, visceral lesions, and primary histology	10	None
van der Linden et al., 2005 [28]	342	Non-operative	Spine	Performance status, visceral lesions, and primary histology	6	Cox proportional hazards
Katagiri et al., 2005 [29]	350	Operative and non-operative	Spine and extraspinal region	Performance status, the number of bony lesions, visceral lesions, primary histology, and prior systemic therapy	8	Cox proportional hazards
Katagiri et al., 2014 [30]	808	Operative and non-operative	Spine and extraspinal region	Performance status, the number of bony lesions, visceral lesions, primary histology, prior systemic therapy, and laboratory data (C-reactive protein, lactate dehydrogenase, albumin, calcium, and bilirubin levels, and platelet count)	10	Cox proportional hazards
Ghori et al., 2015 [38]	307	Operative	Spine	Modified Bauer score, performance status, and albumin level	3	Multivariate logistic regression
Paulino Pereira et al., 2016 [30]	649	Operative	Spine	Performance status, the number of spinal lesions, visceral lesions, primary histology, prior systemic therapy, age, white blood cell count, and hemoglobin level	12	Cox proportional hazards

**Table 2 diagnostics-14-00962-t002:** Features included in conventional prognostic scoring systems.

Feature	Frequency	References
Primary tumor histology and the presence of visceral metastases	8	[23,24,25,26,27,28,29,30,34]
Performance status	6	[23,24,28,29,30,34]
The number of spinal metastases	5	[23,24,25,26,34]
The number of bony metastases	5	[23,24,27,29,30]
Prior systemic therapy	3	[29,30,34]
Serum albumin level	2	[30,38]
Neurologic deficit	2	[23,24]
Pathologic fracture	1	[25]
Age, WBC count, and hemoglobin level	1	[34]
Abnormal C-reactive protein, lactate dehydrogenase, calcium, and bilirubin levels and platelet count	1	[30]

**Table 3 diagnostics-14-00962-t003:** Machine learning-driven prognostic tools.

Study	Number	Treatment	Features	Model
Paulino Pereira et al., 2016 [34]	649	Operative	Performance status, the number of spinal lesions, visceral lesions, primary histology, prior systemic therapy, age, WBC count, and hemoglobin level	Boosting regression
Karhade et al., 2019 [45]	1790	Operative	Performance status, ASA class, albumin level, WBC count, hematocrit, alkaline phosphatase level, and spinal location	Bayes Point Machine
Karhade et al., 2019 [46]	732	Operative	Performance status, visceral lesions, primary histology, BMI, creatinine level, alkaline phosphatase level, albumin level, platelet count, absolute lymphocyte count, hemoglobin level, INR, neutrophil–lymphocyte ratio, and platelet–lymphocyte ratio	Stochastic gradient boosting
Karhade et al., 2022 [53]	3001	Operative and non-operative	Performance status, the number of spinal lesions, visceral lesions, brain lesions, primary histology, albumin level, absolute lymphocyte count, WBC count, and alkaline phosphatase level	Elastic net penalized logistic regression

**Table 4 diagnostics-14-00962-t004:** Features included in machine learning-driven prognostic scoring systems.

Feature	Frequency	References
Performance status	4	[34,45,46,52]
Primary tumor histology and the presence of visceral metastases	3	[34,46,52]
Serum albumin and alkaline phosphatase level	3	[45,46,53]
WBC count	3	[34,45,53]
The number of spinal metastases	2	[34,53]
Absolute lymphocyte count	2	[46,53]
Hemoglobin level	2	[34,46]
Hematocrit, spinal region, and ASA class	1	[45]
Creatinine level, platelet count, neutrophil–lymphocyte ratio, platelet–lymphocyte ratio, and body mass index	1	[46]
Prior systemic therapy and age	1	[34]

## Data Availability

No new data were created or analyzed in this study. Data sharing is not applicable to this article.

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
