# Peer review of "Predictive Modeling for Spinal Metastatic Disease"

_diagnostics, 2024, doi:10.3390/diagnostics14090962_

Round 1
Reviewer 1 Report
Comments and Suggestions for Authors
This is a review article about predictive models of outcomes and survival in patients with spinal metastasis. It is an important topic in the practice of orthopedics as it is required to decide if surgical intervention would do any good in terms of patient morbidity and quality of life.
The authors compared various studies using conventional statistical approaches (Table 1) and also other studies using machine learning methods (Table 2).
Although the review is comprehensive, not too much synthesis of the knowledge is attempted. For example, the authors would like to know :
1. among factors reported in table 1, were there any factors found universal to all studies that were important indicators for prognosis?
These factors would be the key determinants in this area.
The authors may show in a Venn diagram or other suitable presentation method for this purpose.
2. Similarly, for the machine learning identified variables or factor, the same can be done.
3. Finally, the factors found by statistics and machine learning can be compared.
Reviewer 2 Report
Comments and Suggestions for Authors
This review paper discusses the review and success of machine learning methods in accurately predicting mortality in spinal metastatic disease. The scope of the review study covers a rather narrow scientific field. The scope of the paper should encompass a comprehensive exploration of various aspects related to the development, application, and evaluation of predictive models in the context of spinal metastatic disease. The paper is quite inadequate and lacking in terms of being evaluated as a review study with a technical systematic approach. My evaluations and suggestions about the paper are listed below:
1. Abstract should inform about the main objectives and result of the review article (informative abstract).
2. At the end of the introduction section, there should be a paragraph about the novelty of the proposed review paper.
3. The last paragraph of the Introduction section should mention the organization of the rest of the paper. Please include.
4. The approach and systematics of the survey presented at the beginning are in place. However, the article is presented in a disorganized manner and its organization needs to be reconsidered.
5. Paper also does not have a coherent structuring of the topic (such as methodological approaches, chronological order)
6. Listed below are recent research papers with similar content. What are the different aspects of the present study from these studies? What are the different aspects of the present study from these studies? What is your advantage or disadvantage? Why did you do such a study? What is your motivation? What is the contribution of such a study to the literature? In general, these are not given in detail in the paper. So, to attract the reader's attention, you should present the article from a different perspective.
a. Katsos, K., Johnson, S. E., Ibrahim, S., & Bydon, M. (2023). Current applications of machine learning for spinal cord tumors. Life, 13(2), 520
b. Yagi, M., Yamanouchi, K., Fujita, N., Funao, H., & Ebata, S. (2023). Revolutionizing spinal care: Current applications and future directions of artificial intelligence and machine learning. Journal of Clinical Medicine, 12(13), 4188.
c. Kuijten, R. H., Zijlstra, H., Groot, O. Q., & Schwab, J. H. (2023). Artificial Intelligence and Predictive Modeling in Spinal Oncology: A Narrative Review. International Journal of Spine Surgery, 17(S1), S45-S56.
7. Recent literature (especially 2022-2024) should be included. Some studies are listed below. Please include more comprehensive new studies in the paper.
a. Gao, L., Cao, Y., Cao, X., Shi, X., Lei, M., Su, X., & Liu, Y. (2023). Machine learning-based algorithms to predict severe psychological distress among cancer patients with spinal metastatic disease. The Spine Journal, 23(9), 1255-1269..
b. Shi, X., Cui, Y., Wang, S., Pan, Y., Wang, B., & Lei, M. (2024). Development and validation of a web-based artificial intelligence prediction model to assess massive intraoperative blood loss for metastatic spinal disease using machine learning techniques. The Spine Journal, 24(1), 146-160.
c. Wang, D., Sun, Y., Tang, X., Liu, C., & Liu, R. (2023). Deep learning-based magnetic resonance imaging of the spine in the diagnosis and physiological evaluation of spinal metastases. Journal of Bone Oncology, 40, 100483.
d. Zhong, G., Cheng, S., Zhou, M., Xie, J., Xu, Z., Lai, H., ... & Zhang, Y. (2023). External validation of the SORG machine learning algorithms for predicting 90-day and 1-year survival of patients with lung cancer-derived spine metastases: a recent bi-center cohort from China. The Spine Journal, 23(5), 731-738.
e. Motohashi, M., Funauchi, Y., Adachi, T., Fujioka, T., Otaka, N., Kamiko, Y., ... & Sato, S. (2024). A New Deep Learning Algorithm for Detecting Spinal Metastases on Computed Tomography Images. Spine, 49(6), 390-397.
f. Katsos, K., Johnson, S. E., Ibrahim, S., & Bydon, M. (2023). Current applications of machine learning for spinal cord tumors. Life, 13(2), 520.
8. The scope of this review paper should encompass a comprehensive exploration of various aspects related to the development, application, and evaluation of predictive models in the context of spinal metastatic disease. Here's a breakdown of the potential scope:
a. Rationale for Predictive Modeling: Discuss the need for predictive modeling in spinal metastatic disease management, highlighting challenges in prognosis, treatment selection, and patient outcomes prediction.
b. Types of Predictive Models: Describe different types of predictive models used in spinal metastatic disease, such as statistical models, machine learning algorithms, and deep learning approaches.
c. Data Sources and Variables: Explore the sources of data used to develop predictive models, including clinical data, imaging findings, molecular markers, and patient-reported outcomes. Discuss the importance of selecting relevant variables and feature selection techniques.
d. Methodologies for Model Development: Explain the methodologies employed for developing predictive models, including model training, validation techniques, and performance evaluation metrics.
e. Identify limitations and challenges associated with predictive modeling in spinal metastatic disease, such as data availability, model interpretability, and clinical implementation barriers.
Please, summarize the key findings of the review paper, emphasizing the significance of predictive modeling in improving outcomes for patients with spinal metastatic disease and outlining areas for further investigation.
